# Photobiomodulation for Neurodegenerative Diseases: A Scoping Review

**DOI:** 10.3390/ijms25031625

**Published:** 2024-01-28

**Authors:** Qi Shen, Haoyun Guo, Yihua Yan

**Affiliations:** 1MOE Key Laboratory of Laser Life Science, Institute of Laser Life Science, College of Biophotonics, South China Normal University, Guangzhou 510631, China; guohaoyunsl@163.com (H.G.); mshk419@126.com (Y.Y.); 2Guangdong Provincial Key Laboratory of Laser Life Science, College of Biophotonics, South China Normal University, Guangzhou 510631, China

**Keywords:** neurodegenerative diseases, photobiomodulation, Alzheimer’s disease, Parkinson’s disease

## Abstract

Neurodegenerative diseases involve the progressive dysfunction and loss of neurons in the central nervous system and thus present a significant challenge due to the absence of effective therapies for halting or reversing their progression. Based on the characteristics of neurodegenerative diseases such as Alzheimer’s disease (AD) and Parkinson’s disease (PD), which have prolonged incubation periods and protracted courses, exploring non-invasive physical therapy methods is essential for alleviating such diseases and ensuring that patients have an improved quality of life. Photobiomodulation (PBM) uses red and infrared light for therapeutic benefits and functions by stimulating, healing, regenerating, and protecting organizations at risk of injury, degradation, or death. Over the last two decades, PBM has gained widespread recognition as a non-invasive physical therapy method, showing efficacy in pain relief, anti-inflammatory responses, and tissue regeneration. Its application has expanded into the fields of neurology and psychiatry, where extensive research has been conducted. This paper presents a review and evaluation of studies investigating PBM in neurodegenerative diseases, with a specific emphasis on recent applications in AD and PD treatment for both animal and human subjects. Molecular mechanisms related to neuron damage and cognitive impairment are scrutinized, offering valuable insights into PBM’s potential as a non-invasive therapeutic strategy.

## 1. Introduction

Neurodegenerative diseases are a class of disorders characterized by the gradual degeneration and death of nerve cells (neurons) in the central nervous system (CNS) or peripheral nervous system (PNS). This degeneration leads to a progressive decline in cognitive, motor, and/or sensory functions. Neurodegenerative diseases encompass a broad range of conditions, including Alzheimer’s disease (AD), Parkinson’s disease (PD), amyotrophic lateral sclerosis (ALS), Huntington’s disease (HD), frontotemporal dementia (FTD), and prion disease. The global prevalence of neurodegenerative diseases is substantial and continues to increase due to the growing aging population. According to the World Alzheimer Report 2019 [1], approximately 50 million people worldwide have dementia, with AD being the most common cause. PD, ALS, and other neurodegenerative conditions collectively affect millions more across the globe [2,3]. Researchers are actively exploring genetic, environmental, and lifestyle factors that contribute to the increase in neurodegenerative diseases, aiming to develop effective treatments or interventions [4].

However, neurodegeneration is a physiological process that has yet to be completely fully understood. This paper reviews the preclinical and clinical evidence for the neurobiological and behavioral effects of a non-pharmacological physical therapy method, photobiomodulation (PBM) therapy, in brain degeneration and proposes the necessity of more controlled studies.

## 2. Treatment of Neurodegenerative Diseases

Currently, researchers and doctors are actively exploring various treatment methods, including drug therapy, gene therapy, immunotherapy, and cell therapy, to address neurodegenerative diseases.

Drug therapy: For Alzheimer’s disease, drug therapies such as cholinesterase inhibitors (e.g., donepezil) are used to improve cognitive function [5]. Hormone therapy and antidepressants are also employed for symptom management. For PD, dopamine replacement therapy, mainly with levodopa, is the primary treatment, and other medications (e.g., monoamine oxidase type B inhibitors, amantadine, anticholinergics, β-blockers, or dopamine agonists) may be initiated first to avoid levodopa-related motor complications [6]. Deep brain stimulation and medication adjustments are commonly used for symptom relief.

Gene therapy: For Alzheimer’s disease, gene-based therapeutic targets include APP, MAPT, and APOE [7]. For Huntington’s disease, researchers are developing gene-editing techniques targeting the Huntington’s disease gene mutation to alleviate disease symptoms [8].

Immunotherapy: For Alzheimer’s disease, some studies focus on using antibodies or vaccines to clear amyloid plaques in the brain to slow disease progression [9].

Cell therapy: Scientists are actively developing cell-based therapies for diverse neurodegenerative diseases impacting the CNS. Small molecules and growth factors are employed to prompt human pluripotent stem cells (hPSCs) to generate neural progenitor cells (NPCs), resulting in specific lineages situated in distinct regions, including the forebrain, retina, midbrain, spinal cord, and the main cell types affected by various diseases in the entire nervous system [10]. For Parkinson’s disease, the application of stem cell therapy is under investigation, with the aim of replacing damaged dopamine neurons [11]. However, this approach is still in the research phase.

In conclusion, these therapeutic strategies have varying applications and research stages across different neurodegenerative diseases. While there is currently no definitive cure, these approaches hold promise for alleviating symptoms and slowing disease progression, while enhancing the quality of life for patients. In recent years, numerous researchers have suggested non-pharmacological interventions for treating or preventing various neurodegenerative diseases [12,13,14]. Among them, low-level laser therapy or PBM therapy has attracted much attention due to its beneficial brain effects on animals and humans [15,16,17,18,19,20,21].

## 3. Photobiomodulation Therapy

PBM is a method that utilizes low-power light from lasers or light-emitting diodes (LEDs) with wavelengths ranging from red to near-infrared (650–1200 nm) to regulate biological function or induce therapeutic effects. Simply put, it is a light-dependent treatment for various diseases. The energy delivered by this treatment method is low enough to prevent any thermal effects. Several previous studies demonstrated the positive impact of PBM on neurodegenerative disorders, including AD, PD, and different types of epilepsy [22,23,24,25]. Despite the successful outcomes of animal studies, few clinical studies that have shown positive outcomes of PBM against neurodegenerative disease have been conducted. Considering the intricate nature of the pathogenesis of neurodegenerative diseases, methods with broad effects, such as PBM, hold promise. Recently, applications of PBM for the nose and brain have shown to be effective in delaying the progress of the neurodegenerative diseases such as AD and PD. In a recent paper, professor Audrey Valverde et al. found that using PBM therapy that applies specific wavelengths of light onto body tissues can have the effect of reducing the neuropathological and behavioral deficits in AD by controlling hypertension [26]. Research has shown that PBM is applied to target organs at specific wavelengths and optimal fluences or energy density to produce therapeutic effects, without any adverse effects [27,28]. PBM therapy enhances cerebral blood flow, brain energy metabolism, and antioxidant defenses [29,30,31,32]. Additionally, another study group examined the effect of PBM and the application of red to near-infrared light on body tissues, neuroinflammatory responses, and oxidative stress in animal models of neurodegenerative diseases [33]. Neurodegenerative diseases are increasingly prevalent in aging populations worldwide, yet a definitive cure for these conditions remains elusive [34]. Therefore, developing new treatment methods that are safer and more effective in controlling the symptoms of neurodegenerative diseases is an urgent challenge for current research. This review aims to summarize the brain responses to PBM in different preclinical and clinical studies by focusing on mechanisms, neurobiological foundations, and biophysical aspects.

## 4. Mechanisms of PBM on the Brain

PBM refers to the photochemical and photobiological changes occurring at cellular and molecular levels, mediated by photoreceptors and dependent on the absorption of light by cells [35]. Currently, the light sources relevant to PBM mainly encompass blue light, green light, yellow light, red light, and near-infrared light. Target groups within cells associated with these light sources include cytochrome c oxidase, visual pigments, flavins and flavoproteins, porphyrins, nitrogen-containing compounds, and nitrite reductases [36].

Cytochrome c oxidase (CCO), a crucial component of the electron transport chain, absorbs photons through interactions with enzymes [37,38]. These interactions cause alterations in proton gradients, triggering an increase in ATP production [39]. They also influence the regulation of reactive oxygen species (ROS) and activation of transcription factors, thereby initiating cell proliferation and inducing cascading signal responses [40,41]. These actions subsequently result in heightened cell proliferation and migration, particularly in fibroblasts. This regulatory process influences cytokines, growth factors, and inflammatory mediators, consequently averting cell apoptosis through the initiation of anti-apoptotic signals. The absorption spectrum of cytochrome c oxidase primarily focuses on red light (630–670 nm) and the near-infrared range (780–940 nm) [42]. Furthermore, beneficial effects of wavelengths such as 1064, 1068, and 1072 nm have been reported in various studies [43,44,45,46]. Given the elevated levels of light scattering within the 800–850 nm range (the primary absorption peak of CCO), it is conceivable that photons may encounter challenges in penetrating deeper tissue layers when compared to longer wavelengths. Nevertheless, within the spectral range of 1064–1072 nm, CCO exhibits reduced light absorption in comparison to the 800–850 nm range. This reduction in absorption, combined with diminished hemoglobin absorption and reduced light scattering, facilitates the unhindered propagation of light through biological tissues. This phenomenon enables the stimulation of either CCO or ion channels situated within deeper anatomical structures [47,48,49]. Additionally, red light can release various nitric oxide (NO) complexes, including nitrosylmyoglobin, S-nitrosothiols, nitrosylhemoglobin, and nitrosyliron compounds, subsequently affecting mitochondrial function [50].

On the other hand, blue and green light can activate visual pigments, which are G-protein-coupled receptors that can activate transient receptor potential channels [51]. The influx of calcium ions following this activation stimulates calcium-dependent kinase II, phosphorylates the cyclic AMP (cAMP) response element-binding protein in the cell nucleus, and further promotes a series of gene transcription expressions. Flavin mononucleotide exists in complex I of the electron transport chain, where it can perceive blue light to provide activation energy, facilitating the catalysis of oxygen reduction to superoxide [52]. Complex II, another flavin-containing complex, can sense blue light and causes a series of biological effects [53].

## 5. The Level of Light Penetration into the Brain

As researchers pay more and more attention to PBM treatment of the brain, some tissue optics laboratories have studied the penetration ability of different wavelengths of light through the scalp and skull as well as the penetration depth of different wavelengths of light in brain parenchyma. This is an interesting issue to consider, as the power density threshold (mW/cm^2^) required to cause biological effects on the brain is currently unclear. The ability of light to penetrate biological tissues is closely related to a variety of optical parameters, such as light scattering, absorption, reflection, refraction, wavelength, energy, power, irradiance, radiation exposure (influence), exposure time, continuous wave and pulse wave, irradiation plan, exposure area, distance between the light source and the target tissue, etc. Based on the understanding and analysis of the differences of individual physiology and anatomical structure of tissue samples, as well as the optical properties of various brain regions, it is helpful to select the best optical parameters in PBM treatment to achieve the best penetration effect.

The general principle of light tissue interaction within the 650–1200 nm wavelength optical window most commonly used in PBM is that shorter-wavelength light has lower tissue penetration compared to higher-wavelength light. Ando et al. observed that the penetration rate of an 810 nm laser through the scalp and skull of BALB/c mice is 6.26% [54]. Lapchak et al. compared the transmittance of 810 nm light in four different species of skulls and found that the transmittance of mice was 40%, while that of rats was 21%, rabbits 11.3%, and human skulls only 4.2% [55]. Salehpour et al. examined light penetration at different wavelengths and found that approximately 23% of the 630 nm laser and 40% of the 810 nm laser reached the surface of the prefrontal cortex of Wistar rats [56]. Zhang et al. observed that a red laser (635 nm) is capable of penetrating through approximately 30% the scalp and skull of C57BL/6 mice [57]. By comparing the penetration ability of light with different wavelengths (660 nm, 808 nm, and 940 nm) in the human cadaveric brain, Tedord et al. found that the penetration ability was the best when using 808 nm wavelength, reaching a depth of 2.52 mm in brain tissue. The brain tissue has a small water absorption peak near the 940 nm wavelength [58]. Beyond the 900 nm wavelength, higher wavelengths lead to higher head penetration, thereby delivering effective levels of energy to deeper brain regions. The comparison of three representative PBM wavelengths, 660 nm, 810 nm, and 1064 nm, for transporting photons to the human head using a Monte Carlo simulation showed that 1064 nm is the optimal wavelength due to its reduced light scattering [59]. Although other photoreceptors (such as water) also absorb photons with higher wavelengths within this range, 1064 nm has less light scattering due to its longer wavelength, thus having deeper penetration ability for the human head. Therefore, in order to achieve the optimal PBM of the human brain, the trade-off between the absorption of light by CCO and the depth of light penetration must be considered [18].

The debate regarding whether the penetration depth of lasers into tissue is indeed greater than that of LED lights remains unsettled in current scientific discourse. Some researchers have suggested that, in contrast to laser devices generating coherent beams with a narrow focus for penetration, non-coherent LED devices exhibit spatial divergence, limited to superficial layers of tissue [60,61]. Conversely, for LEDs (830 nm) and lasers (810 nm), there appears to be no significant disparity in light penetration through human cranial bones [62]. Further laboratory studies are necessary to elucidate the precise process of light transmission through human skull tissue.

## 6. Specific Applications of PBM Therapy for Neurodegenerative Diseases

Considering the significant dependence of the brain on mitochondrial activity, it is unsurprising that PBM has undergone extensive testing for the treatment of various brain diseases [63]. PBM activates multiple signaling pathways, including those mediated by ROS, resulting in an upregulation of antioxidant defenses [64]. Anti-apoptotic and pro-survival signals are also triggered [65]. Furthermore, there are two other critical impacts on the ability to shift mitochondrial respiration from glycolysis to oxidative phosphorylation. Firstly, it mobilizes stem cells from hypoxic niches, allowing them to migrate to sites of injury, where they can facilitate repair. Secondly, alterations in mitochondria can shift the phenotype of macrophages and microglial cells from pro-inflammatory M1 states to anti-inflammatory and phagocytic M2 states [66]. In the brain, the upregulation of neurotrophic factors, such as the brain-derived neurotrophic factor (BDNF), stimulates neurogenesis and fosters synaptic formation and neural plasticity [67,68,69]. Therefore, PBM can regulate several signal transduction and metabolic pathways dependent on neuronal survival. For example, it can improve the survival of neurons, promote neurogenesis, reduce cerebral inflammatory reaction, enhance cerebral blood flow, and maintain mitochondrial homeostasis, so as to improve the response of neurons to cell damage in the process of neurodegenerative disease progression. This review shows that a variety of biological signal transduction and metabolic regulation pathways are affected by PBM in typical neurodegenerative diseases. Combined with the optimal application of specific optical parameters, the molecular response of such cells will be pushed in the right direction (Figure 1).

### 6.1. Alzheimer’s Disease

#### 6.1.1. Reduction in the Burden of β-Amyloid Plaques

PBM effectively reduces the burden of amyloid plaques by stimulating the phagocytosis of protein aggregates via microglia or by regulating the level of the enzymes involved in the production and degradation pathways of β-amyloid (Aβ) peptides. PBM promotes microglia phagocytosis by activating Rac1, thus avoiding inflammation-induced cytotoxicity [70]. Professor Wei et al. indicated that the 1070 nm light can induce morphological changes in microglia and increase their colocalization with Aβ to reduce Aβ load [71]. Aβ peptides originate from the sequential proteolytic cleavage of the amyloid precursor protein (APP) by β-secretase, also known as β-site APP cleaving enzyme 1 (BACE1), as well as the γ-secretase complex, comprising presenilin 1 (PS1), nicastrin, and Pen-2. This proteolytic process significantly contributes to the pathological mechanisms underlying Alzheimer’s disease [72]. Nonetheless, the generation of Aβ can be avoided through a nonamyloidogenic pathway involving α-secretase, primarily represented by a disintegrin and metalloproteinase domain-containing protein 10 (ADAM10), in conjunction with the γ-secretase complex [73]. PBM activated the cAMP/PKA/SIRT1 pathway, thereby upregulating ADAM10 and downregulating BACE1, ultimately altering the hydrolysis process of APP. Additionally, the upregulation of ADAM10 and downregulation of BACE1 by PBM were dependent on SIRT1-coupled retinoic acid receptor β and PPARγ coactivator 1α deacetylation [57]. In addition, PBM visibly increased the level of Aβ-degrading enzyme and insulin degrading enzyme (IDE), but not neutral endopeptidase (NEP), in 5xFAD mice [74]. The study indicated that after internalizing Aβ_42_ in human neuroblastoma cells (SH-EP), irradiation with a 670 nm laser can significantly reduce the accumulation of Aβ_42_ in the cells [75].

#### 6.1.2. Neuroinflammation

Glial cells represent a substantial category of cells in the nervous tissue distinct from neurons. They are broadly distributed in both the central and peripheral nervous systems and primarily encompass astrocytes, oligodendrocytes (collectively referred to as macroglia), and microglia [76]. Microglial cells display two distinct phenotypes in response to inflammation. In the absence of pathogens, microglia remain quiescent. However, upon the detection of pathological hallmarks of AD, microglia become activated, releasing inflammatory mediators, and participate in the clearance of Aβ plaques. Activated microglia manifest in either a pro-inflammatory or an anti-inflammatory state [77]. M1 microglia assume a pro-inflammatory state, continuously releasing inflammatory factors such as ROS and NO, which, through excessive pruning of neuronal synapses and demyelination, ultimately result in disrupted information transmission and even cell death [78]. In contrast, M2 microglia adopt an anti-inflammatory state, capable of phagocytosing Aβ plaques, providing trophic factors, promoting neural network reconstruction, and facilitating brain repair in response to injury [79,80].

In various neurodegenerative diseases, including AD, PBM plays a crucial role in modulating central nervous system damage mediated by microglial cells. Irradiating primary microglial cells and microglia-like BV-2 cells with an 808 nm LED at different doses, it was observed that at higher doses (4–30 J/cm^2^), microglial cells predominantly transitioned to the M1 phenotype, whereas at lower doses (0.2–10 J/cm^2^), they primarily shifted towards the M2 phenotype [77]. Using a 633 nm laser, Song et al. [70] explored the effect of PBM on the neurotoxicity induced by lipopolysaccharide (LPS)-activated microglial cells. The results revealed that PBM can inhibit the activation of LPS-stimulated microglial cells by activating the non-receptor tyrosine kinase Src. This inhibition reduced the neurotoxic effects caused by microglial cells, enhanced their phagocytic capabilities, and consequently facilitated the clearance of Aβ aggregates. Furthermore, Professor Qu’s research group utilized 808 nm laser irradiation on primary microglial cells from the mouse hippocampus, which were activated by Aβ_1–42_ oligomers [81]. The results indicated that in M1-type activated microglial cells induced by Aβ oligomers, PBM effectively induced a metabolic shift from glycolysis to mitochondrial activity. This shift led to the reduced secretion of pro-inflammatory factors and enhanced phagocytic function, transitioning the microglial cells towards the M2 phenotype. Simultaneously, in a co-culture model of neurons and activated microglial cells, PBM was found to effectively reduce neuronal apoptosis by modulating the levels of ROS generated by Aβ-activated microglial cells.

Furthermore, similar to microglial cells, astrocytes also exhibit two states, pro-inflammatory (A1) and anti-inflammatory (A2), during CNS inflammation. A1 astrocytes release pro-inflammatory factors such as NO and ROS, which exert toxicity in glial cells, rendering them phagocytosis-impaired and incapable of clearing Aβ plaques [82]. A2 astrocytes release ATP, γ-aminobutyric acid (GABA), neurotrophic factors, and cytokines, promoting neuronal proliferation, differentiation, and synaptic repair [83]. Early studies revealed that 632.8 nm laser pretreatment can suppress superoxide dismutase activity in Aβ_1–42_-induced astrocytes, reducing the expression of interleukin-1β (IL-1β) and inducible nitric oxide synthase (iNOS), thereby exerting an anti-neuroinflammatory effect [84]. Silveira et al. [85] found that irradiating C6 astroglioma cells with a 660 nm laser led to increased respiratory chain complex I activity and hexokinase activity, resulting in alterations in cytoplasmic and mitochondrial redox states, enhanced ATP production, and significantly improved C6 cell mitochondrial activity. Yoon et al. [86] discovered that treating primary astrocytes from SD rats with a 660 nm LED increased the co-expression of GFAP and bromodeoxyuridine (BrdU)/Ki67, promoting specific proliferation. Additionally, the co-expression of Oct4 and GFAP decreased, while Nestin and ALDH1L1 expression increased, suggesting that a 660 nm LED can facilitate astrocyte differentiation. These findings indicate that PBM can modify astrocyte activity, promoting their proliferation or differentiation, thereby ameliorating the pathological state of the CNS.

For the in vivo test, Blivet et al. [57] applied a transcranial pulse PBM using a combination of red and NIR light, causing a reduction in hippocampal inflammation markers TNF-α, IL-6, and IL-1β. Simultaneously, it decreased the activation of astrocytes and microglial cells, leading to the restoration of spatial working memory in the Y-maze test for mice. Professor Wei Xunbin et al. used continuous LED devices emitting light in the range from 1040 nm to 1090 nm for transcranial PBM in APP/PS1 mice. This treatment significantly improved spatial learning and memory in the Morris water maze test for AD mice and reduced the area of Aβ plaques in the hippocampal region [87]. In 2021, the same team conducted transcranial PBM on AD model mice using 10 Hz and 40 Hz 1070 nm near-infrared light. The results indicate that both 10 Hz and 40 Hz 1070 nm near-infrared light can reduce the production of M1-like microglial cells, alleviating the inflammatory response in AD model mice [71]. In 2016, the results of Li-Huei Tsai et al. showed that a 40 Hz LED (white light) induced gamma oscillations in the visual cortex of the brain, leading to morphological changes in microglial cells in AD mice and subsequently enhancing amyloid protein phagocytosis to reduce the occurrence of AD [88]. In 2019, their team explored the effectiveness of combined visual and auditory stimulation at 40 Hz. They discovered that this combined stimulation could promote the clustering of microglial cells around Aβ plaques, leading to a widespread reduction in amyloid proteins and phosphorylated Tau proteins in multiple brain regions [89]. This, in turn, improved spatial memory and cognitive abilities in AD mice. In the same year, the research group investigated the mechanism of long-term 40 Hz light therapy. The results reveal that long-term 40 Hz light therapy induced gamma oscillations in multiple regions of the brain, including the visual cortex, hippocampus, and frontal cortex. This led to changes in the expression of genes related to neurons, synaptic damage, and microglial cell morphology, mitigating neuron damage and microglial cell inflammatory responses, improving synaptic function, enhancing neuroprotection, and ultimately enhancing cognitive function in AD mice [90]. Furthermore, PBM promoted the level of IL-3Rα in microglial cells and IL-3 in astrocytes, facilitating the recruitment of microglial cells around amyloid plaques [91].

#### 6.1.3. Neurogenesis

The CNS comprises neural stem cells (NSCs), which produce various neurons and glia and seem to be very suitable as alternative therapies for various cell loss diseases [92]. Under the pathological conditions of neurodegenerative diseases, especially during the development of AD, adult hippocampal neurogenesis (AHN) is inhibited, which reduces the stability of the neural circuit and leads to a decrease in the production of new neurons in the dentate gyrus (DG), making it difficult for AD to recover from learning/memory dysfunction. Professor Da Xing’s research group found that PBM facilitated adult hippocampal neurogenesis in AD mice. Both in vitro and in vivo, PBM triggered the activation of latent transforming growth factor-β1 (LTGFβ1) during AHN induction. This activation subsequently enhanced the differentiation of NSCs in the hippocampus of APP/PS1 mice, promoting their transition into newly generated neurons [93]. Numerous findings have shown that the immune system is closely related to the synaptic plasticity, inflammation, and progression of AD [94,95,96,97,98,99]. However, few studies have improved the niche microenvironment of NSCs in the brain of AD patients by regulating the function of nonparenchymal immune cells, in order to promote AHN. Therefore, in 2022, for the first time, their research team found that PBM regulates peripheral CD4^+^ T lymphocytes, which contributes to promoting adult hippocampal neurogenesis and alleviating AD cognitive impairment [100]. Furthermore, a descriptive report on dementia care from Ivan V Maksimovich et al. showed that 48 patients suffering from AD received transcatheter intracerebral laser PBM through catheterization to stimulate neurogenesis and tissue structure recovery, improving cerebral blood flow and leading to a reduction in dementia and enhancement of cognitive function, an effect that can last for a long time [101].

#### 6.1.4. Apoptosis

Cell apoptosis stands as one of the pathophysiological mechanisms in normal brain aging and neurodegenerative diseases. The phenomenon of PBM anti-apoptotic properties was initially observed in skeletal muscle satellite cells by Shefer et al. [102]. The study highlighted that laser irradiation at a wavelength of 632.8 nm could diminish the levels of p53, p21, and Bax, while concurrently elevating Bcl-2 levels 24 h post-irradiation, thus safeguarding skeletal muscle satellite cells from apoptosis. Additionally, LED light at 640 nm inhibited apoptosis in PC12 cells induced by Aβ toxicity, as evidenced 24 h post-irradiation [103]. Light at 670 nm (LED) [104] and 810 nm (laser) [105] substantially prevented apoptosis by decreasing pro-apoptotic factors as well as by inhibiting Caspase-3. One plausible explanation for the neuroprotective effect of this light could be attributed to the initial mechanism of light absorption by chromophores within the mitochondrial inner membrane enzymes, leading to an enhancement in mitochondrial membrane potential (MMP) [106]. Furthermore, Zhang and colleagues demonstrated that low-dose laser irradiation at 632.8 nm effectively reversed apoptosis in PC12 cells by reducing the mRNA Bax/Bcl-xl ratio through the activation of the PKC pathway [107]. Laser irradiation at 632.8 nm notably suppressed the activation of glycogen synthase kinase 3β, Bax, and Caspase-3, effectively averting staurosporine-induced apoptosis [108].

#### 6.1.5. Cerebral Blood Flow and Neurotrophic Factors

Nitric oxide (NO) is a potent vasodilator that may be released from its binding sites through a photodissociation process within the respiratory chain when undergoing PBM. Based on preclinical research, PBM has the capacity to elevate the levels of neuronal NO, enlarge blood vessel diameter, and enhance cerebral blood flow (CBF) [109,110]. Consequently, it is plausible to consider that PBM therapy targeted at specific brain regions may have the potential to influence regional CBF [111]. Transcranial PBM utilizing red and NIR wavelengths resulted in an elevation of cerebral oxygen consumption in both naive rats and AβPP mice [112]. Additionally, in the latest clinical studies conducted by Hanli Liu et al., enhancements in cerebral oxygenation and hemodynamics were noted during and after transcranial laser irradiation at 1064 nm [30,47].

Among the various members within the neurotrophic factors family, known as neurotrophins, research has shifted towards exploring the stimulating impacts of PBM on BDNF, nerve growth factor (NGF), and glial cell-derived neurotrophic factor (GDNF). The upregulation of neurotrophins such as BDNF and NGF could explain the observed enhancements in neurogenesis and synaptogenesis [113]. Increasing BDNF expression may contribute to the attenuation of cortical dendrite atrophy in the CNS during the progression of AD [69]. In this context, it has been proposed that PBM (632.8 nm) can rescue dendritic atrophy by activating the ERK/CREB/BDNF pathway [69]. In similar research employing the same laser wavelength (632.8 nm), PBM triggered IP3 receptor activation, leading to elevated Ca^2+^ levels. Subsequently, this activation promoted the ERK/CREB pathway, ultimately culminating in heightened expression of BDNF [114]. For in vivo experiments, a coherent laser light at 670 nm significantly elevated BDNF in the cortex of rats [104].

Currently, despite the substantial financial investments made in the development of therapeutic drugs for AD, significant breakthroughs in its treatment efficacy remain elusive. However, the therapeutic efficacy and safety of PBM have been validated in numerous preclinical and clinical studies (Table 1), suggesting that PBM may emerge as a promising alternative therapeutic approach for neurodegenerative diseases. By leveraging advantages such as a non-invasive nature and lack of side effects, PBM can serve as an adjunctive therapy to existing effective treatments and even evolve into a preventive therapeutic modality. In summary, PBM holds great promise in the treatment and prevention of cognitive impairments.

### 6.2. Parkinson’s Disease

Parkinson’s disease is a multifactorial, multisystem disorder characterized by the loss of dopaminergic neurons in the substantia nigra pars compacta (SNc) of the brain [137]. The depletion of dopamine (DA) predominantly influences motor function, giving rise to symptoms such as bradykinesia, muscle rigidity, resting tremors, and postural instability. However, it also manifests in various other signs and symptoms affecting emotions, cognition, the digestive system, and the sense of smell. The death of dopaminergic neurons is attributed to the accumulation of Lewy bodies, which are aggregates of α-synuclein, within the cells, although this theory remains somewhat controversial. The exact cause of PD is not entirely understood. A genetic component is observed in only around 15% of PD patients, with the most common mutations occurring in genes such as leucine-rich repeat kinase 2, glucocerebrosidase, and α-synuclein [138,139].

Numerous studies have presented promising outcomes regarding the application of PBM in animal models of Parkinson’s disease (refer to Table 2). A recent review of animal evidence has indicated that human trials are warranted based on the encouraging findings [140]. Photobiomodulation has demonstrated efficacy in pre-treating and safeguarding animals, including non-human primates, from the effects of toxin (MPTP)-induced Parkinson’s disease models. This protection extends to mitigating induced PD signs and preserving substantia nigra neurons [141,142]. Previous studies show that PBM restored DA levels in toxin (MPP+)-exposed cells by increasing the expression of the dopamine transport protein VMAT2 and tyrosine hydroxylase, the enzyme responsible for DA synthesis. The regulation of VMAT2 expression occurred via the ERK/CREB pathway, a critical regulator in maintaining, releasing, and expressing monoamines like dopamine and serotonin in the central nervous system monoaminergic neurons, thereby underscoring the neuroprotective effects of PBM [143,144,145]. Changes in mitochondrial dynamics, such as mitochondrial fission, fragmentation, and functional loss, are commonly observed in PD [146]. Post-treatment with PBM preserved healthy mitochondrial dynamics and suppressed the mitochondrial fragmentation in CA1 neurons. This treatment reduced the detrimental activity of dynamin-related protein 1 (Drp1) GTPase, mitochondrial fission factor (Mff), and mitochondrial fission 1 protein (Fis1) and balanced mitochondrial targeting fission and fusion protein in global cerebral ischemia [147]. Moreover, there are reports indicating that twice-daily LED treatment (at 670 nm) markedly reduced the count of striatal and cortical neurons undergoing apoptosis induced by exposure to rotenone and MPP+ [148]. Findings from a mouse model of PD reveal that the application of photobiomodulation at 670 nm to distal tissues (excluding the head) exerts a notable ex vivo neuroprotective effect by mitigating the loss of midbrain dopaminergic neurons [149]. The pathogenesis of PD is associated with abnormalities in the SNc, a midbrain structure located at an approximate depth of 80–100 mm from the coronal suture, beneath the dura mater. Research suggests that near-infrared light may not penetrate the human brain effectively, with a depth of penetration not exceeding 20 mm beyond the cortical surface [150]. This limitation is deemed noteworthy in the context of transcranial photobiomodulation therapy for Parkinson’s disease.

## 7. Conclusions and Future Perspectives

A substantial body of research on PBM has been published, including investigations into its mechanisms of action and methods for assessing therapeutic outcomes, providing a strong evidence-based foundation for future clinical trials. However, PBM has a long journey ahead before becoming a widely accessible therapeutic approach. Over the past decade, PBM has been extensively employed in drug development for various diseases, such as AD, demonstrating favorable therapeutic effects. Nevertheless, several unresolved issues have been identified. Currently, its mechanisms of action and dose–response relationships, especially the dose–response relationship in AD treatment, remain unclear. More comprehensive and in-depth clinical trials are required to elucidate the optimal treatment modalities and dosage parameters to adopt in clinical practice and to refine the clinical assessment system. Simultaneously, there is a need to develop portable PBM treatment devices tailored for neurodegenerative diseases, offering a promising new direction for clinical treatment and a feasible solution to alleviate the burdens of disease management and healthcare costs for patients.

## Figures and Tables

**Figure 1 ijms-25-01625-f001:**
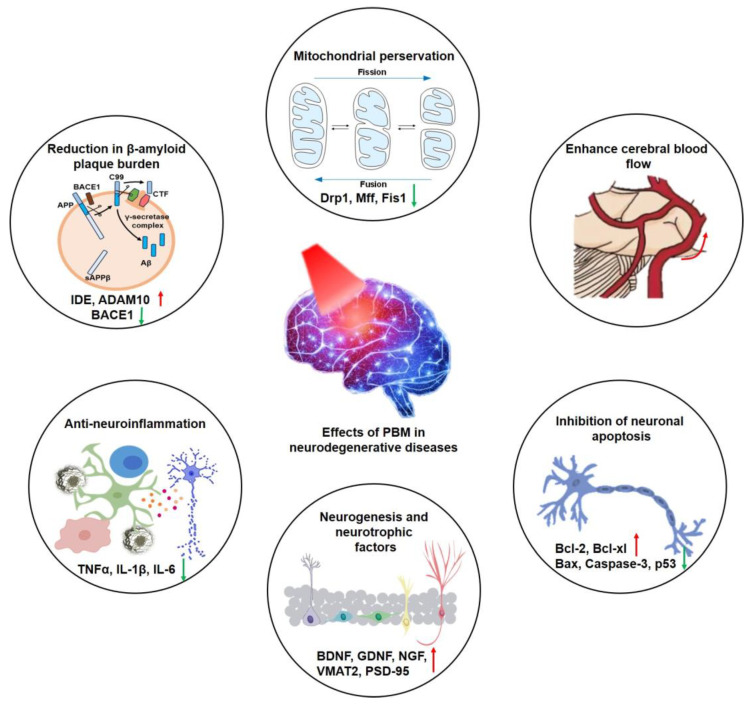
The effects of PBM in neurodegenerative diseases.

**Table 1 ijms-25-01625-t001:** Summary of in vivo studies on the effects of photobiomodulation therapy in Alzheimer’s disease.

	Index	Irradiation Parameters					Irradiation Site	Target Molecule or Mechanism of Action	Application Object	References
Disease		Light Source	Wavelength	Continuous/Frequency	Power Density (mW/cm^2^)	Dose Density (J/cm^2^)				
Alzheimer’s disease		LED	473 nm	CW; 40 Hz	/	52.63	Eyes/Head	Inflammation, biological rhythm, increased SOD activities, decreased AChE activity	Mouse, Human	[88,89,90,115,116]
		LED	500 nm	CW	0.23	/	Eyes	Sleep	Human	[117]
		LED	610–670 nm	CW; 40 Hz	20–70	2–9	Head/Nasal cavity/Eyes	Aβ oligomers, tau proteins, increased CCO activity, neurogenesis, biological rhythm	Mouse, Human	[29,57,69,74,84,93,100,118,119,120,121,122,123,124]
		Fiber laser	808/810 nm	CW	25–566	3–68	Head/Nasal cavity/Eyes	Aβ oligomers, mitochondrial function, blood flow	Mouse, Rat, Human	[21,32,91,112,125]
		Fiber laser	808/810 nm	10–100 Hz	13–100	3–25	Head/Eyes	mitochondrial function, inflammation, sleep	Rat, Human	[32,126,127,128]
		Fiber laser, LED	1040–1267 nm	CW; 10, 40, 600 Hz	5–250	4.5–32	Head/ Whole body	Aβ oligomers, heat shock protein	Mouse, Human	[71,87,129,130,131,132,133,134,135,136]

**Table 2 ijms-25-01625-t002:** Summary of in vivo studies on the effects of photobiomodulation therapy in Parkinson’s disease.

	Index	Irradiation Parameters					Irradiation Site	Target Molecule or Mechanism of Action	Application Object	References
Disease		Light Source	Wavelength	Continuous/Frequency	Power Density (mW/cm^2^)	Dose Density (J/cm^2^)				
Parkinson’s disease		Laser	405 nm	CW	/	/	Head	Decreased activity of monoamine oxidase-B and acetylcholinesterase	Rat	[151]
		Fiber laser, LED	627–675 nm	CW	14–50	0.5–80	Head/Whole body/Intracranial implantation	Dopaminergic neurons, blood vessels, inflammation	Mouse, Rat, Monkey	[141,144,145,149,150,152,153,154,155,156,157,158,159,160,161,162,163,164,165,166]
		Semiconductor laser, LED	808–810 nm	CW	5–25	0.5–2.5	Head	Dopaminergic neurons, mitochondrial function	Mouse, Rat, Human	[167,168,169]
		Semiconductor laser	904 nm	50 Hz	/	/	Head/Mouth	/	Human	[170]

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
