# Peer review of "Photobiomodulation for Neurodegenerative Diseases: A Scoping Review"

_ijms, 2024, doi:10.3390/ijms25031625_

Round 1

Reviewer 1 Report

Comments and Suggestions for Authors

Low power laser therapy or photobiomodulation is a non-invasive treatment that utilizes red or infrared light to stimulate biological functions. In the current review paper, the authors discuss the application of PBM to neurogenerative diseases in general and more specifically to Parkinson and Alzheimer’s disease. The authors have performed a detailed and comprehensive literature search and their findings are presented in a very detailed and precised manner.

I believe that the paper under review should be accepted for publication in its current form with only a few minor changes, most probably typographic mistakes:

1.       Add spaces before the brackets with bibliography references, e.g. line 37, Alzheimer Report 2019[1] to Alzheimer Report 2019 [1].

2.       Line 94, replace “in this paper” with “In a recent paper”.

3.       Line 118, CCO absorbs photons it does not generate them. Please correct.

4.       Line 170, Delete “And” before “Zhang”.

5.       Line 183-184, correct the phrase “there is a trade-off between the absorption of light and the depth of light penetration by CCO” as follows “there is a trade-off between the absorption of light by CCO and the depth of light penetration”.

6.       Line 341, there is no need for capitalization of A after the word furthermore

Reviewer 2 Report

Comments and Suggestions for Authors

I appreciate the opportunity to review the manuscript entitled “Photobiomodulation for the neurodegenerative diseases” for International Journal of Molecular Sciences. In this review process, I sought to thoroughly analyze every aspect of the work, aiming to contribute to its improvement in terms of quality and scientific impact.

While this work addresses a relevant topic and makes a significant scientific contribution, some important considerations are necessary:

1.    The title must contain information about the study design. What type of review?

2.    The manuscript, despite being well written, does not have the necessary characteristics to be considered a systematic review, as currently written it is compatible with a non-systematic narrative review. Comprehensively review the entire manuscript, with clear guidance on its objectives, so that it can be considered for publication.

3.    The term preventative physiotherapeutic modality should be replaced by preventive therapeutic modality, since it is a treatment that is not restricted to physiotherapists only.

4.    English language review is recommended.

Comments on the Quality of English Language

 English language review is recommended. 

Round 2

Reviewer 2 Report

Comments and Suggestions for Authors

The file submitted as the revised manuscript is the same as the previous version. I believe there might have been an error. I look forward to receiving the manuscript with the requested changes. Thank you.

Comments on the Quality of English Language

The revised file sent is the same as the previous version. I am awaiting the correct file for a new evaluation.